# Endovascular Approaches to the Cavernous Sinus in the Setting of Dural Arteriovenous Fistula

**DOI:** 10.3390/brainsci10080554

**Published:** 2020-08-14

**Authors:** Justin Dye, Gary Duckwiler, Nestor Gonzalez, Naoki Kaneko, Robert Goldberg, Daniel Rootman, Reza Jahan, Satoshi Tateshima, Viktor Szeder

**Affiliations:** 1Department of Neurosurgery, Loma Linda University, Loma Linda, CA 92354, USA; 2Division of Interventional Neuroradiology, Department of Radiological Sciences, David Geffen School of Medicine, University of California, Los Angeles, CA 90095, USA; gduckwiler@mednet.ucla.edu (G.D.); NKaneko@mednet.ucla.edu (N.K.); RJahan@mednet.ucla.edu (R.J.); STateshima@mednet.ucla.edu (S.T.); VSzeder@mednet.ucla.edu (V.S.); 3Department of Neurosurgery, Cedars-Sinai Medical Center, Los Angeles, CA 90048, USA; nestor.gonzalez@cshs.org; 4Department of Ophthalmology, David Geffen School of Medicine, University of California, Los Angeles, CA 90095, USA; Goldberg@jsei.ucla.edu (R.G.); rootman@jsei.ucla.edu (D.R.)

**Keywords:** cavernous sinus, carotid-cavernous fistula, neuroendovascular techniques

## Abstract

Dural arteriovenous fistulas involving the cavernous sinus can lead to orbital pain, vision loss and, in the setting of associated cortical venous reflux, intracranial hemorrhage. The treatment of dural arteriovenous fistulas has primarily become the role of the endovascular surgeon. The venous anatomy surrounding the cavernous sinus and venous sinus thrombosis that is often associated with these fistulas contributes to the complexity of these interventions. The current report gives a detailed description of the alternate endovascular routes to the cavernous sinus based on a single center’s experience as well as a literature review supporting each approach. A comprehensive understanding of the anatomy and approaches to the cavernous sinus available to the endovascular surgeon is vital to the successful treatment of this condition.

## 1. Introduction

The cavernous sinuses (CS) are dural venous channels located on either side of the body of the sphenoid bone. The CS has four walls, three of which consist of two layers of dura (outer meningeal and inner periostic). The medial wall, the only with a single layer of dura, makes up the lateral limit of the sella. The lateral wall contains cranial nerves (CN) III, IV, and V and lies just medial to the temporal pole. The superior wall, or roof of the CS, stretches from the diaphragma sellae medially to the anterior clinoid process laterally. The posterior wall contributes to the dura of the clivus [1]. The cavernous segment of the internal carotid artery (ICA) runs within the CS with CN VI running along its inferior and lateral border.

The major tributaries to the CS are the superior ophthalmic vein (SOV), inferior ophthalmic vein, superficial middle cerebral vein (SMCV), sphenoparietal (sphenobasal) sinus, and the contralateral sinus through the intercavernous (circular) sinus. The major egress of the CS includes the superior petrosal sinus (SPS), inferior petrosal sinus (IPS), emissary veins to the pterygoid plexus, clival plexus to the posterior fossa and paraspinal venous plexus, and the contralateral sinus through the intercavernous sinus.

In 1985, Barrow et al. classified carotid cavernous fistulas (CCF) into direct (Type A) or indirect (Type B, C, or D) fistulas [2]. A direct CCF, commonly caused by trauma, is a direct connection between the cavernous segment of the ICA and the CS. Indirect carotid cavernous fistulas (dural CCF) are a type of dural arteriovenous fistula (DAVF) that occurs when an abnormal connection forms between the CS and meningeal branches of the external carotid artery (ECA), meningeal branches of the internal carotid artery (ICA), or the ICA itself.

The presentation, work up, and management options for CCFs are well published [3,4,5]. Indications for treatment include progressively worsening symptoms, vision loss, and/or cortical venous reflux (CVR). For DAVFs in general, the goal of treatment is to block the fistulous connection. Many times, direct CCFs can be treated through a transarterial approach with access to the CS across the tear in the cavernous segment of the carotid artery [3,6]. The fistula can then be obliterated with a balloon or stent protecting the cavernous ICA. Secondary to multiple small arterial feeders, which may supply cranial nerves, dural CCFs are often treated with a transvenous approach to the CS followed by coil or liquid embolic embolization of the CS itself. [3,4,7]. Complex venous anatomy and sinus thrombosis, often associated with dural CCFs, make a detailed knowledge of CS anatomy essential for safe and effective endovascular treatment.

The current report focuses on dural CCFs and the multiple transvenous routes to the CS available to the endovascular surgeon.

## 2. Inferior Petrosal Sinus Via the Internal Jugular Vein

The inferior petrosal sinus is a common route to the CS in the setting of dural CCFs [3,4,7,8]. The IPS originates in the posterior/inferior aspect of the CS. It travels posterolaterally within the inferior petrosal sulcus and passes through the anterior portion of the jugular foramen. The IPS then terminates in the anteromedial aspect of the jugular bulb. The junction of the IPS with the IJV has variable anatomy and was classified into four types by Shiu et al. in 1968 [9]. In their description, type I anatomy is a single IPS with a small caliber or no connection to the vertebral venous plexus (VVP). Type II anatomy is a single IPS with a large caliber connection to the VVP. In type III anatomy, a network of small veins make up the IPS and form the connection with the IJV. In type IV anatomy, there is no connection between the IPS and the IJV. More recently, Miller et al. classified the venous anatomy of 135 patients undergoing bilateral inferior petrosal sinus sampling (IPSS) [10]. The authors found true type IV anatomy in only 1% of patients and were successful in bilateral IPSS in 99% of patients. For the IPS approach via the IJV, the authors of the current article prefer to place a 5 Fr sheath in the left common femoral artery (CFA) and a 5 Fr catheter in the common carotid artery (CCA), to be used for diagnostic angiograms and roadmaps. A 6 Fr 23 cm sheath is then placed in the right common femoral vein (CFV), for operator comfort, and a 6 Fr guide catheter is navigated into the internal jugular vein (IJV). Next, using roadmapping technique a curved microwire directed anteriorly and medially is used to select the IPS. The IPS is then navigated superiorly and medially until the tip of the microcatheter is advanced into the posterior CS. An intermediate catheter, such as a Distal Access Catheter (DAC) (Stryker Neurovascular, Fremont, CA, USA), can be used for extra support if necessary. In the case of a thrombosed IPS, access to the CS is still possible by “drilling” through the thrombus with a microsystem. A microwire is navigated in the direction of the expected IPS anatomy. A torque device is used to twist the wire repeatedly as gentle forward pressure is applied. The microcatheter is intermittently advanced to provide enough support for the wire to continue along its path. It is also possible to direct the wire into the clival venous plexus, which runs parallel to the IPS, and still reach the CS through this alternate pathway. The microwire drilling technique should be used with caution as perforation into the subarachnoid space can result in intracranial hemorrhage. Secondary to this risk, the authors recommend avoiding stiff microsystems during this approach (usually nothing stiffer than a 014” standard microwire).

In general, the authors most commonly use a standard 0.010” or 0.014” microwire with a compatible microcatheter. A softer, more torqueable microwire is usually chosen first, followed by stiffer wires as needed. Microwires larger than 0.014” are rarely necessary. In longer approaches, such as with the middle temporal vein or angular vein to the SOV, a longer and softer microcatheter is selected such as the MicroVention Headway Duo 167 cm (Aliso Viejo, CA, USA) or the Medtronic Marathon 165 cm (Irvine, CA, USA). When torqueability and support are needed in these longer approaches, usually a 0.010 wire is used. When the anatomy is particularly tortuous a softer wire is usually chosen such as the Medtronic Mirage 0.008” (Irvine, CA, USA) or the Balt Hybrid 0.007”/0.012” (Irvine, CA, USA).

When it is present and patent, the IPS provides a direct route to the CS. However, when this route is not available, alternatives do exist.

## 3. Superior Ophthalmic Vein Via the Angular Vein

Another common route to the CS is through the SOV, which is often dilated in dural CCFs. Transfemoral access to the SOV can be achieved by catheterization of the angular vein, which communicates with the SOV via the nasofrontal vein. The frontal and supraorbital veins join to form the angular vein which runs inferiorly in an oblique fashion, lateral to the root of the nose, down to the inferior margin of the orbit where it continues as the anterior facial vein. The anterior facial vein runs deep to the facial artery, superficial to the masseter muscle and joins the retromandibular vein to form the common facial vein, which drains directly into the IJV. The common facial vein can be selected with the curved tip of a micro- or guidewire directed anteriorly at approximately the midcervical segment of the IJV (near the angle of the mandible) (Figure 1). In some cases, the retromandibular vein can be selected from the external jugular vein (EJV). The angular vein is then navigated with a microsystem and roadmapping technique as it courses superiorly and medially. The nasofrontal vein, which lies near the medial/superior orbit, is then selected after obtaining a localized magnified roadmap. Next, the SOV is selected and is followed posteriorly and inferiorly where it communicates with the superior/anterior portion of the CS (Figure 2). When planning this approach, careful consideration of the length and compatibility of the coaxial systems is critical. This route can be quite long and tortuous requiring a tri-axial system with an intermediate catheter for support. For example, a shorter intermediate catheter allows for the maximum amount of usable microcatheter length to navigate the tortuous angles, but if it is too short, you may not be able to reach your target. The authors commonly use an 80 cm long sheath placed in the IJV; a 115–120 cm intermediate catheter is placed in the angular vein, followed by a microsystem to access the CS.

Another technique to keep in mind when using this approach is the ability to “milk” the microcatheter in these superficial subcutaneous facial veins. With one finger, gentle pressure can be applied to the skin directly over the microcatheter in a sweeping fashion towards the eye. Alternatively, physically manipulating the skin and tissue can change the angle of the vein and allow the microcatheter to advance or even direct the catheter into a desired branch. This technique can be useful when attempting to navigate a difficult angle or gain more support by removing the slack in the system.

Yu et al. reported endovascular embolization of 74 CCFs in 71 patients using a transvenous approach, with an overall success rate of 86.5% [11]. The IPS was used to access the CS in most patients. However, in 10 patients, the IPS was not accessible, and the authors approached the CS through the SOV via the angular vein (in one patient the SOV was selected via the middle temporal vein). Prior to adopting this transfacial approach, the authors had a technical success rate of 71.6%, while after employing this technique, their success rate improved to 100%. Berkman et al. described direct percutaneous puncture of the facial vein using ultrasound guidance to treat a dural CCF [12]. In their report, multiple attempts at transfemoral embolization had failed, and the facial vein ended in a web of small veins making it impossible to traverse with a microsystem. Therefore, the authors performed a percutaneous puncture of the infraorbital segment of the facial vein using ultrasound guidance and a 21-gauge micropuncture needle. A 0.018-inch wire was then passed into the facial vein, and the needle was exchanged for a 4 Fr sheath. Using roadmapping technique, a microsystem was then navigated into the angular vein, through the SOV, and subsequently into the CS. After embolization with coils and n-BCA glue, transfemoral angiograms showed complete resolution of the fistula. The authors of the current report have used a similar technique for direct percutaneous puncture of the angular vein itself.

The angular vein/SOV approach is not only useful in cases of IPS thrombosis but also when the fistula site is located within the anterior CS and does not communicate with the posterior CS. This is a common occurrence secondary to compartmentalization of the CS. The IPS approach may only provide access to the posterior CS, while the SOV allows the operator to gain access to an anteriorly located fistula site.

## 4. Superior Ophthalmic Vein Via the Middle Temporal Vein

As described in the study by Yu et al., the SOV can also be accessed via the middle temporal vein [11]. To reach the middle temporal vein, the retromandibular vein must first be accessed. In many patients, the retromandibular vein can be selected from the EJV. In some patients, the common facial vein is first selected at the midcervical segment of the IJV, as described above. In the later approach, the retromandibular vein is selected off the common facial vein by directing a curved wire postero-laterally, as opposed to the anterior facial vein, which is selected by directing the wire antero-medially. The retromandibular vein (AKA posterior facial vein) is formed by the junction of the superficial temporal and maxillary veins. It travels inferiorly within the parotid gland, deep to the facial nerve, before it divides into a posterior branch, which communicates with the EJV and an anterior branch, which combines with the anterior facial vein to form the common facial vein. The retromandibular vein is navigated superiorly, past the anteriorly directed maxillary vein, where it continues as the posterior and superiorly directed superficial temporal vein. Next, the middle temporal vein is selected as it travels anteriorly and medially where it communicates with the SOV (Figure 1 and Figure 3). Since the superficial temporal and middle temporal veins are superficial scalp vessels, this is another approach in which it is possible to manually manipulate the skin in order to change the angle of the vein or “milk” the microcatheter underneath the skin (Figure 4). This may help to traverse difficult tortuous segments in the vein. In the case of angular vein thrombosis or hypoplasia, the middle temporal vein serves as a viable alternative to access the SOV and subsequently the anterior CS.

## 5. Superior Petrosal Sinus Via the Transverse Sinus

In 2002, Mounayer, et al. described the endovascular treatment of a dural CCF utilizing the SPS for access to the CS [13]. In their patient, the IPS could not be catheterized, and approaching the SOV via the facial vein was not possible. Therefore, a 5 Fr sheath was placed in the IJV, and a 5 Fr guiding catheter was placed in the transverse sinus (TS). Then, the SPS was catheterized with a microsystem, which allowed retrograde access to the CS. After the placement of detachable coils in the CS, final angiograms showed no evidence of residual fistula. The SPS originates in the posterior/superior portion of the CS at the petrous apex, travels posterior and lateral in the superior petrosal sulcus, and drains into the distal transverse sinus. The acute angle between the SPS and the TS can make this a challenging approach. Approaching the SPS from the contralateral TS across the torcula gives a much more favorable angle and should be considered when attempting this approach. Given the anatomic course of this approach, there is a risk of subarachnoid hemorrhage in the event of vessel breach. For this reason, it is recommended that soft microwires be utilized, the wire be shaped with a gentle curve, and forceful movements should be avoided.

## 6. Pterygoid Plexus Via the Maxillary Vein

As one of the major egresses of the CS, the pterygoid plexus can be used as an access route in the setting of dural CCFs. In 1998, Jahan et al. described the endovascular treatment of a right-sided dural CCF via the contralateral pterygoid plexus [14]. The venous drainage in this case was directed contralaterally through the intercavernous sinus into the left CS and then inferiorly into the left pterygoid plexus. Secondary to thrombosis of the bilateral SOVs and a lack of filling of the bilateral IPSs, the authors chose to approach the right CS from the left pterygoid plexus. The right CS was then accessed through the intercavernous sinus, and coil placement within the right CS resulted in complete resolution of the fistula. The pterygoid plexus lies between the temporalis and the lateral pterygoid muscles. This venous plexus can be accessed by first selecting the common facial vein off the midcervical segment of the IJV. The retromandibular vein is then selected with a curved wire directed postero-laterally. Next, the anteromedially directed maxillary vein is selected. The maxillary vein is a direct tributary of the pterygoid plexus which is navigated with a microsystem using a localized magnified roadmap. The pterygoid plexus communicates with the anterior/inferior CS through small unnamed veins of the foramen of Vesalius (medial to the foramen ovale) and emissary veins of the foramen lacerum. The pterygoid plexus is often an enlarged emissary vein, which can be navigated medially and superiorly to reach the CS. Most of this approach is extracranial and may have a lower risk of subarachnoid hemorrhage than the “in-out-in” technique of trying to navigate a thrombosed or stenotic IPS. However, the disadvantage is the technical difficulty of traversing the tight net of vessels that make up the pterygoid plexus, and therefore, it is generally reserved for cases of bilateral IPS and SOV thrombosis or absence.

## 7. Inferior Petrooccipital Vein Via the Internal Jugular Vein

Kurata et al. described an alternative endovascular approach to the CS via the inferior petrooccipital vein (IPOV) [15]. The IPOV was first named by Trolard in 1868 when he discovered a small vein that was distinct from the IPS [16]. Katsuta et al. referred to the IPOV as the inferior petroclival vein. They described it as a small vein that may become dilated in the setting of DAVFs and serve as an alternative route to the CS [17]. The IPOV originates from the clival (basilar) plexus, courses extracranially through the petrooccipital suture, and often contributes to the anterior condylar confluence (ACC) (Figure 5). It can be accessed through the IJV by first selecting the ACC from the jugular bulb. The IPOV is then selected by directing a microwire directly medial from the ACC (more medial than for selection of the IPS). The IPOV is then navigated superiorly as it runs roughly parallel and deep to the IPS. The basilar plexus is then navigated with a localized magnified roadmap before the posterior CS is accessed. The IPOV is often mistaken for the IPS given their parallel routes. However, in the case of IPS thrombosis the IPOV may remain patent, or even be dilated, and allow access to the CS.

## 8. Intercavernous Connections (Intercavernous Sinus and Clival Plexus)

The intercavernous sinus (aka circular sinus) is both a major egress as well as a major tributary to the CS. This venous pathway has an anterior limb, which runs in front of the pituitary stalk, and a posterior limb, running behind the pituitary stalk. The anterior limb is generally larger than the posterior limb, and together with the two CSs, they form the circular sinus. Depending on the venous anatomy, the anterior limb, posterior limb, or both may be absent.

The clival plexus (aka basilar plexus) is a network of veins located between the two layers of the clival dura and drains into the anterior vertebral venous plexus. While the clival plexus primarily serves as an egress of the CS, because of its multiple interconnections, it also provides a pathway between the left and right CS.

In the setting of CCFs, the venous drainage may be primarily through contralateral pathways. Therefore, the best endovascular approach to a left sided CCF may be through the right IPS. Once the contralateral CS has been accessed, the ipsilateral CS can be accessed by crossing the intercavernous sinus, or more commonly through the clival plexus. The intercavernous connection is selected with a curved microwire directed either anteriorly or posteriorly. The microwire is then traversed across the intercavernous connection and, if necessary, looped within the opposite CS to provide support for the microcatheter to follow (Figure 6). This approach gives the operator the option of using virtually any of the described approaches on the contralateral side of the fistula site.

## 9. Percutaneous Transorbital Puncture of the Cavernous Sinus, Inferior Ophthalmic Vein, or Superior Ophthalmic Vein

When transfemoral access to the fistula site is not possible, direct percutaneous transorbital puncture of the CS, SOV, or IOV can be an effective alternative. Kurata et al. described direct percutaneous puncture of the extraconal portion of the SOV in three patients with traumatic CCFs in which transarterial and IPS approaches failed [18]. In their approach, the extraconal portion of the SOV is cannulated with an 18-gauge sheathed needle inserted into the upper portion of the medial orbital angle. The needle is guided by transfemoral angiographic roadmapping technique. The anterior apsidal vein located between the first and second segments (intraconal portion) of the SOV is used as a landmark. More commonly reported is direct puncture of the IOV. The IOV begins as a network of veins along the anterior/medial wall of the orbit. The IOV runs posteriorly along the lower half of the orbit, above the inferior rectus muscle, until it divides into two branches. The inferior branch passes through the inferior orbital fissure and contributes to the pterygoid plexus. The superior branch passes through the superior orbital fissure (SOF) and either enters the anterior CS directly or, more commonly, joins the SOV before draining into the CS. White et al. described their series of eight patients with CCFs treated by direct transorbital puncture of the IOV or CS and subsequent coil embolization [19]. All eight patients experienced complete obliteration of their CCF in a single procedure, and there were no permanent periprocedural complications. In this approach, transfemoral access is first obtained to investigate the fistula as well as to provide transarterial diagnostic angiograms. Once it is discovered that the traditional venous pathways are not accessible, the ipsilateral orbit is prepped and draped in the usual sterile fashion. A 21-gauge needle and a 4 French micropuncture kit are used for percutaneous access. The 21-gauge spinal needle, attached to aspiration tubing, is inserted into the inferolateral aspect of the orbit and gradually advanced along the orbital floor. Under live fluoroscopic guidance, the tip of the needle is directed towards the superior orbital fissure (SOF). The SOF is separated from the optic canal by placing the AP tube in an oblique 45-degree projection, while keeping the floor of the orbit flat. Care is taken to not damage the optic nerve within the optic canal. While the needle is advanced, gentle aspiration is applied to the tubing with a 5 mL syringe until venous return is obtained. The tubing is then removed, and the micropuncture wire (0.018 gauge) is inserted through the needle. Next, the needle is exchanged for a 4 French introducer sheath (alternatively the needle can be left in place and direct injections through the needle can be performed). A venogram is performed through the sheath to confirm positioning within the IOV or the CS itself. Care must be taken to perform only gentle injections in these settings to avoid applying increased pressure in what is already an over pressurized system as this can lead to vessel rupture and significant morbidity. Next, using roadmapping technique a microcatheter is advanced coaxially over a microwire into the CS (Figure 7). Once the embolization is complete, a diagnostic angiogram is performed through the arterial catheter to confirm resolution of the fistula. The microcatheter and sheath are then removed, and gentle pressure is applied to the puncture site. A dressing is generally not necessary. This approach is particularly useful in fistulas located within the anterior CS which have prominent drainage into the ophthalmic veins. Additionally, the direct percutaneous access to the CS avoids the need for surgical exposure of the ophthalmic veins. It is important to note, however, that perforation of an orbital vein carries higher consequences than perforation of a facial (angular) vein. Therefore, this approach should be reserved for patient′s in which lower risk approaches are not possible. It is recommended that ophthalmology be available for emergent optic nerve and/or globe decompression if necessary.

## 10. Surgical Exposure of the SOV or IOV

An alternative that should be considered for anteriorly located dural CCFs with venous drainage through dilated ophthalmic veins is surgical exposure of the SOV or IOV. Surgical exposure of the SOV and the IOV have been previously described [20,21,22,23,24]. Surgical exposure of the SOV is more commonly performed and is described below. Just as in the direct percutaneous approach, transfemoral access is first obtained to investigate the fistula as well as to provide transarterial diagnostic angiograms. Once this is done, an ophthalmologist makes a 15 mm curvilinear incision just below the eyebrow. Using blunt dissection, the SOV is identified, isolated, and secured just below the superior orbital rim at the level of the trochlea. Ultrasound may be helpful in locating the SOV. Systemic heparin is avoided to promote thrombosis of the CS and reduce the risk of hemorrhage. Once the SOV is exposed, a 21-gauge needle is used to puncture the vein within the middle third of the visualized segment. Next, the micropuncture wire is introduced through the needle and guided into the CS under live fluoroscopy. In general, the SOV runs medial to lateral as it courses towards the SOF. However, if the wire does not pass easily in this direction, a lateral to medial approach can be tried as anatomic variations or loops in the vein can exist. Next, the needle is exchanged for the 4 Fr dilator from a standard micropuncture kit (Cook Medical, Bloomington, IN, USA) and is advanced coaxially approximately 2–5 cm. A standard angiocath IV catheter, which is shorter than the 4 Fr dilator, is an alternative if the microwire does not have enough “landing room” prior to a tight turn. Silk sutures are used to secure the sheath to the vein in order to prevent losing access during the procedure. A venogram is then performed through the sheath to confirm positioning within the SOV (Figure 8). The inner diameter of the dilator is 0.038”, which will accept up to a size 17 microcatheter, which is then advanced over a microwire into the CS using roadmapping technique. A venogram through the microcathter is performed prior to embolization with coils and/or embolic glue material. Once the embolization is complete, a diagnostic angiogram is performed through the arterial catheter to confirm resolution of the fistula. The microcatheter and sheath are then removed, and the SOV, which is no longer under arterial pressure, is typically sutured or cauterized on either side of the puncture site. In some situations where orbital venous drainage is very tenuous, maintaining patency of outlet veins can be attempted without cauterization to prevent stagnation or thrombosis. Balanced with the risk of bleeding, consideration should be given to postoperative anticoagulation with initiation of a heparin drip to maintain venous drainage. The wound is closed in layers, and the skin is re-approximated cosmetically. Prior to attempting this approach, non-invasive cerebral vascular imaging should be reviewed carefully to evaluate the SOV anatomy. A small, atretic SOV is a contraindication to this technique. However, a thrombosed SOV with a large diameter is not necessarily a contraindication and may be able to be traversed by the previously described “drilling” technique. Surgical exposure of the SOV or IOV allows for direct visualization and immobility of the arterialized vessel, which may carry a lower risk of vessel rupture than percutaneous puncture.

## 11. Surgical Exposure of the Cavernous Sinus

Despite the multiple routes to the CS available to the endovascular surgeon, in some cases, the venous anatomy and location of the fistula site make treatment with endovascular techniques alone inadequate. In these cases, open microsurgical exposure of the CS combined with endovascular embolization may be an option. Staged microsurgical obliteration of residual DAVFs after incomplete endovascular embolization is well published [4,25,26]. However, simultaneous open surgery and endovascular embolization of CCFs are less commonly reported. Krisht et al. reported the treatment of a dural CCF in which the lateral wall of the CS was surgically exposed and dissected prior to cannulation and coil embolization [27]. Geurrero et al. reported a similar case in which a pretemporal, extradural approach to the lateral CS was utilized for treatment of a dural CCF. Once the lateral wall was exposed, the CS was cannulated under ultrasound guidance prior to embolization of the fistula with coils [28]. We have reported the treatment of a paracavernous venous plexus fistula with an open surgical approach combined with endovascular embolization [29]. Through a pterional craniotomy, an intradural pretemporal approach to the lateral wall of the cavernous sinus was performed. Using a preoperative CT angiogram and intraoperative neuronavigation, the exact location of the fistula site within the paracavernous venous plexus was identified. Next, the tip of a spinal needle, registered to and guided by neuronavigation, was introduced into the fistula. An angiogram through the spinal needle was then performed to confirm positioning within the fistula sac without filling of the ICA (Figure 9). After injection of Onyx embolic material into the fistula, intraoperative transfemoral angiogram confirmed complete obliteration of the fistula. Secondary to the risks involved with open surgery, this type of combined approach is generally reserved for fistulas that have failed embolization by traditional endovascular routes. However, as in the above report, DAVFs with fistula sites in paracavernous locations can make treatment by traditional transvenous routes technically impossible. It is in these cases that combined open surgical and endovascular embolization can be useful.

## 12. Transarterial Via the Cavernous Segment of the ICA (Direct CCF) or Via ICA/ECA Meningeal Branches (Dural CCF)

In the setting of direct CCFs, the tear in the cavernous segment of the ICA provides an endovascular route to the CS. A primary transarterial approach can often be used to embolize these fistulas. Prior to attempting a primary embolization of the fistula, a complete angiogram should be completed, and if adequate cross-filling is identified, a balloon test occlusion (BTO) is recommended. If the patient remains neurologically unchanged, the most effective treatment is often complete ICA occlusion. It is important to remember that the carotid occlusion should extend across the fistula site both proximally and distally to prevent retrograde filling of the fistula from collateral circulation. In the case of failed BTO, the fistula must be treated while maintaining patency of the ICA. In this case, the authors often use a two-catheter system with a microcatheter in the cavernous sinus for embolization and a Scepter balloon catheter (MicroVention, Tustin, CA, USA) to protect the ICA. Using roadmapping technique, a microsystem is then navigated through the tear in the cavernous ICA into the affected CS. An occlusion balloon, or a stent, is placed within the cavernous ICA across the fistula site to protect the ICA during embolization. Coils, Onyx (Covidien, Irvine, CA, USA), or a combination of both are then used to obliterate the fistula. Alternatively, in some cases of direct CCF, a combined transarterial and transvenous approach is the safest and most effective way to embolize the fistula. Figure 10 demonstrates a right sided direct CCF with venous drainage through the intercavernous sinus into a dilated left CS. In this case, the authors first navigated a Hyperform balloon (Covidien, Irvine, CA, USA) across the tear in the right ICA. Next, a microsystem was navigated through the left IPS, into the left CS, and subsequently into the intercavernous sinus. With the balloon inflated to prevent coil migration into the right ICA, the intercavernous sinus was embolized with coils, effectively obliterating the fistula. Final angiograms showed no evidence of residual fistula, and the patient had complete resolution of her preoperative symptoms.

For dural CCFs, a transarterial approach with embolization of the dural ICA or ECA feeders can be an effective treatment strategy. However, navigating multiple small arterial feeders can be difficult and time consuming. Furthermore, given the added risk of cranial nerve deficit with occlusion of mengingeal arteries, the authors prefer the transvenous approach for most dural CCFs and use the transarterial approach sparingly.

## 13. Discussion

The key to approach selection lies in careful study of initial diagnostic angiography. In particular the venous phase is reviewed looking at the connections of the internal and external jugular veins and their respective venous outlet anatomy. Characterization of the connections of the SOV to the facial venous system is of critical importance. The absence of any viable endovascular route may require more direct access to the cavernous sinus.

While this work represents a detailed review of the commonly used endovascular approaches to the CS, it is not intended to be comprehensive. Other, approaches have been described including direct transovale access to the CS. Gil et al. described percutaneous puncture of a CCF that was refractory to standard approaches through the foramen ovale in order to completely obliterate the fistula [30]. Ghosh et al. described a patient who underwent image-guided burr hole placement to allow catheterization of the Sylvian vein and subsequently access the CS and then embolize the patient’s CCF [31]. Endoscopic-assisted transsphenoidal access to the CS for CCF treatment has also been described [32]. These approaches, and more, highlight the creativity of the endovascular surgeon and the multitude of options available.

The transvenous approach to the CS is an effective management strategy for the treatment of dural CCFs. The venous anatomy associated with the cavernous sinus can be complex and highly variable, with hypoplasia and venous thrombosis commonly seen [1,33]. CCF mimics, such as fistulas of the paracavernous sinus, clival plexus, laterocavernous sinus, and dura of the sphenoid wing can further complicate treatment of these DAVFs [29,34]. A detailed knowledge of the multiple pathways to the CS, as well as the surrounding venous anatomy, can be the difference between successful fistula obliteration and a failed attempt.

Many benign dural CCFs will resolve spontaneously with time, and conservative management should be discussed as a treatment option in the absence of worsening vision, increasing intraocular pressure, or CVR [35]. However, waiting too long to intervene can lead to venous outflow obstruction and increase the technical difficulty, and risks, of treatment.

Once the decision to intervene has been made, it is important to keep in mind the goals of therapy, as well as the priority of each goal. The highest risk of death or major disability in a patient with a CCF is intracranial hemorrhage secondary to venous hypertension. For this reason, the authors prefer to disconnect the site of CVR first before addressing orbital symptoms. For example, in a CCF with reflux into both the SMCV and SOV, the authors will first block the venous outlet into the SMCV in the posterior CS (with coils and/or liquid embolic material) and then reposition the microcather into the anterior CS to block the reflux into the SOV. Remember that all injections should be done gently especially within orbital veins or the CS itself in order to avoid over pressurizing an abnormally high-pressure system, which can lead to vessel rupture with intraorbital and/or intracranial hemorrhage. Above all, the fistula site itself must be completely obliterated to avoid redirecting venous outflow into alternate pathways, which can turn a symptomatic CCF with reflux only into the SOV into a life-threatening fistula with reflux into cortical veins.

The evaluation and management of patients with a CCF has largely become the responsibility of the endovascular surgeon. When intervention is indicated in a patient with a dural CCF, a transvenous approach through the IPS is commonly used. However, in the absence of a patent IPS multiple alternative routes do exist, and an understanding of and experience with each approach can greatly enhance the technical success of these procedures.

## Figures and Tables

**Figure 1 brainsci-10-00554-f001:**
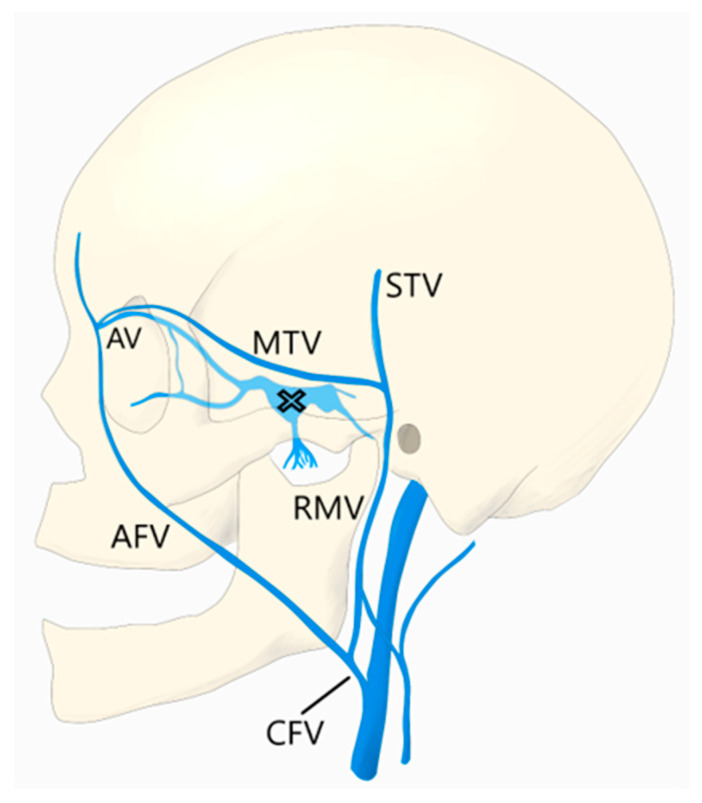
The common facial vein (CVF), anterior facial vein (AFV), angular vein (AV), retromandibular vein (RMV), middle temporal vein (MTV), superficial temporal vein (STV), and cavernous sinus (X) are demonstrated.

**Figure 2 brainsci-10-00554-f002:**
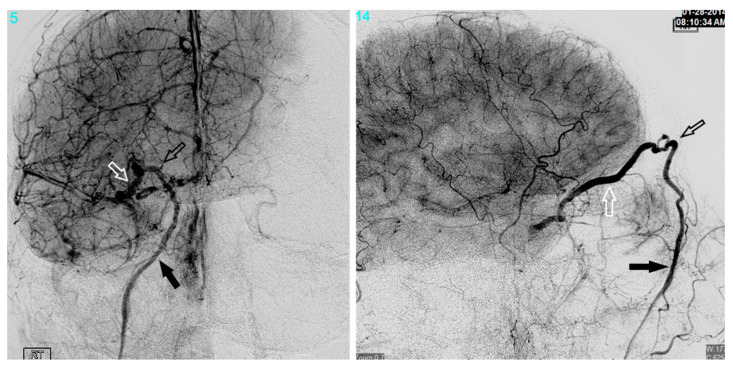
The anterior facial vein (solid arrow), angular vein (black arrow), and superior ophthalmic veins (white arrow) are shown.

**Figure 3 brainsci-10-00554-f003:**
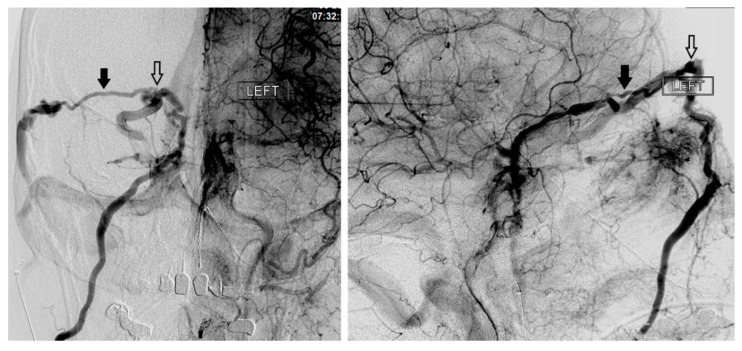
The middle temporal vein (solid arrow) and its connection (open arrow) with the SOV are demonstrated.

**Figure 4 brainsci-10-00554-f004:**
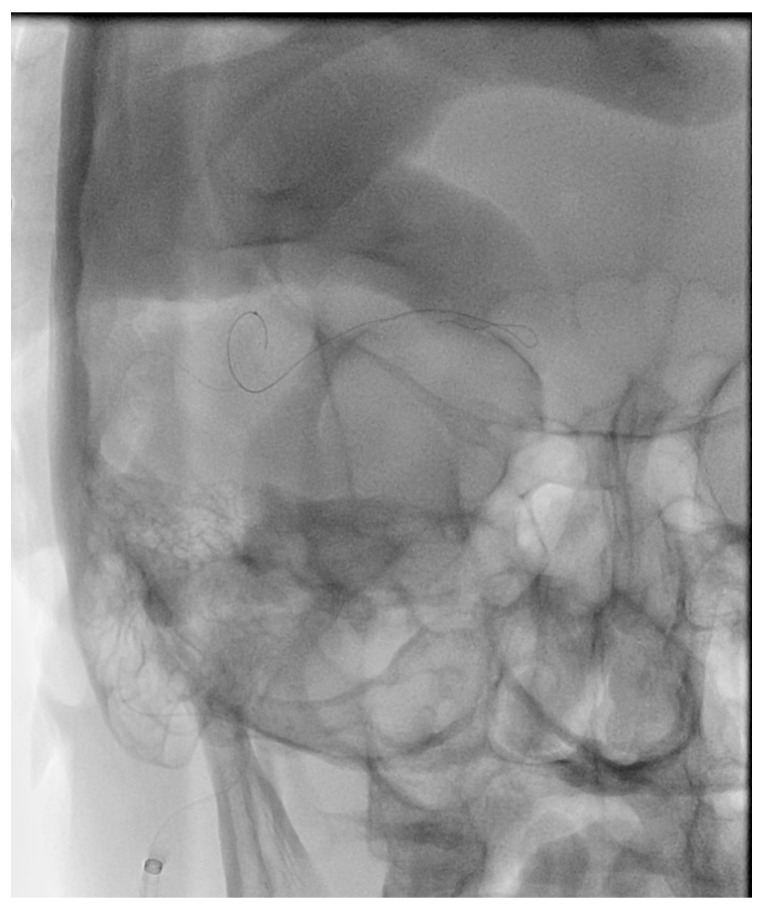
An example of “milking” the microcatheter along the middle temporal vein is shown.

**Figure 5 brainsci-10-00554-f005:**
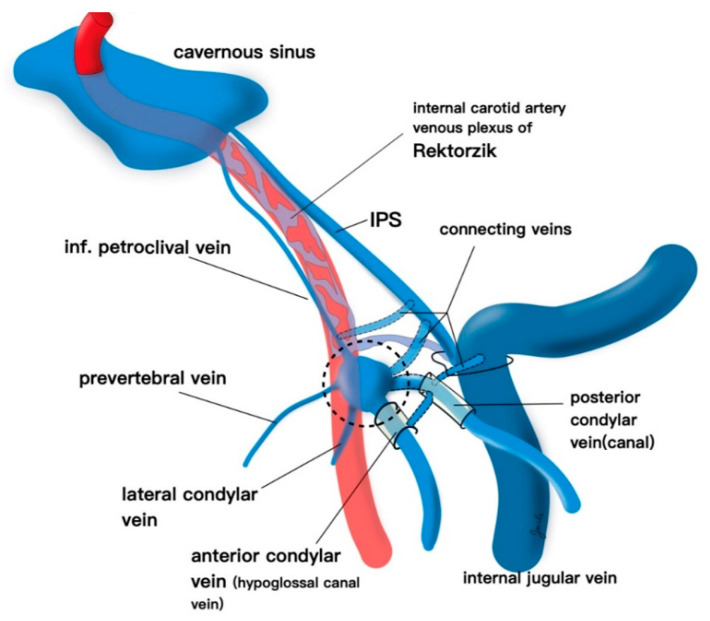
The anterior condylar vein, inferior petroclival (petroocipital) vein, inferior petrosal sinus (IPS), and cavernous sinus are demonstrated.

**Figure 6 brainsci-10-00554-f006:**
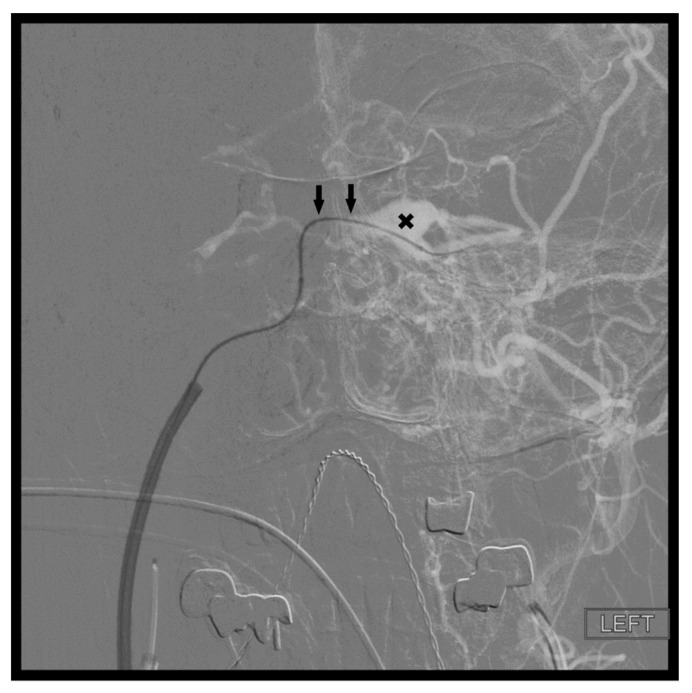
Demonstrates access of the left cavernous sinus (**X**) through an intercavernous approach (arrows).

**Figure 7 brainsci-10-00554-f007:**
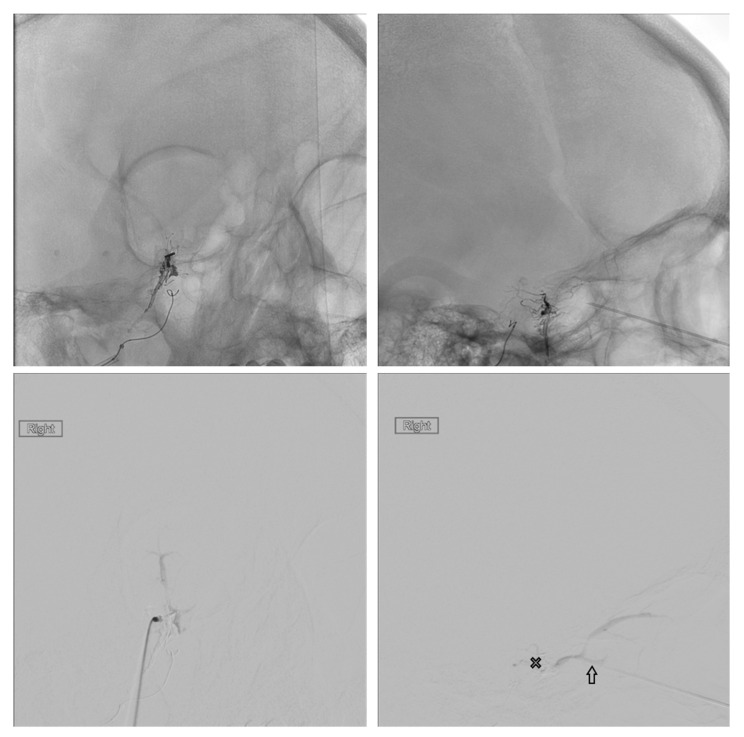
Demonstrates percutaneous transorbital puncture of the cavernous sinus (X) via the inferior ophthalmic vein (arrow).

**Figure 8 brainsci-10-00554-f008:**
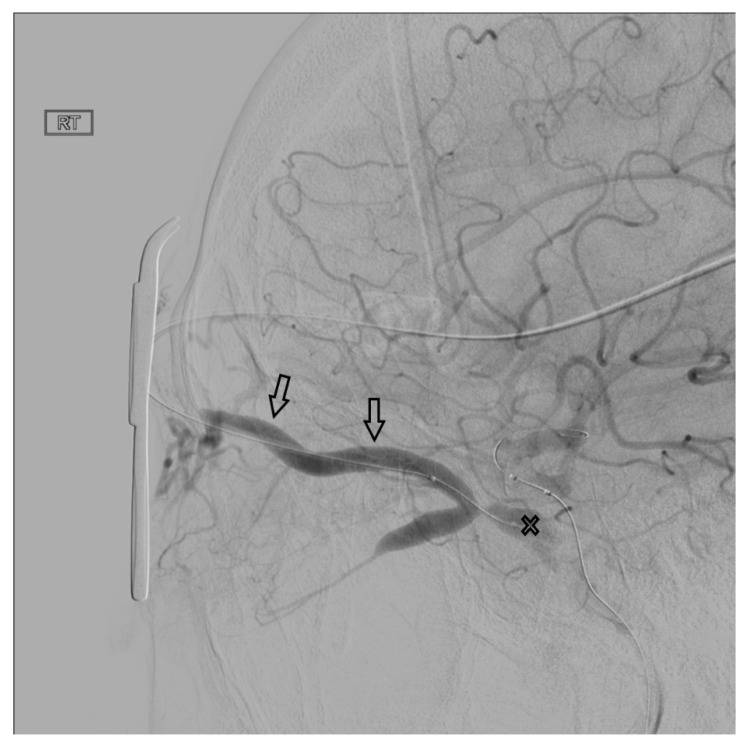
Demonstrates access to the cavernous sinus (X) via surgical exposure and cannulation of the SOV (arrows).

**Figure 9 brainsci-10-00554-f009:**
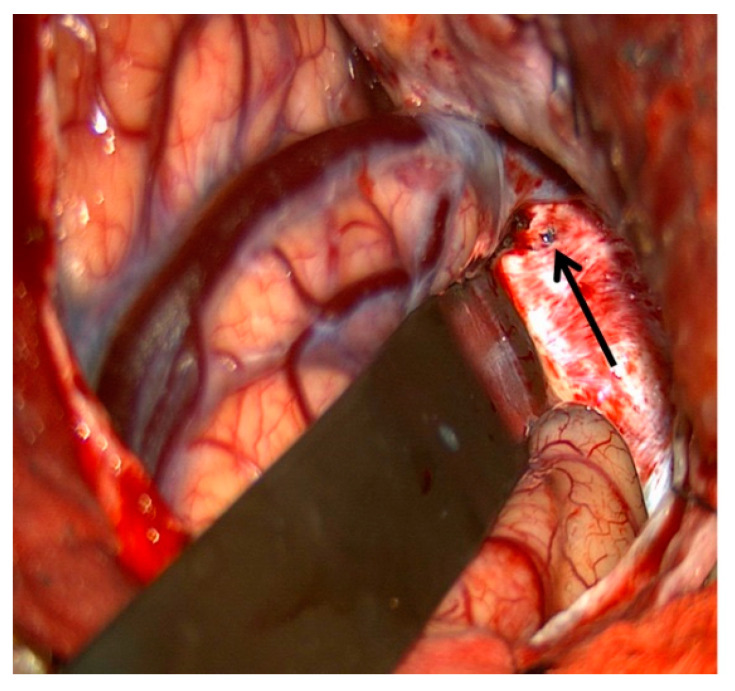
Demonstrates surgical exposure of the lateral wall of the right cavernous sinus via a pre-temporal approach (left). The arrow shows the site of spinal needle puncture in the lateral wall of the cavernous sinus.

**Figure 10 brainsci-10-00554-f010:**
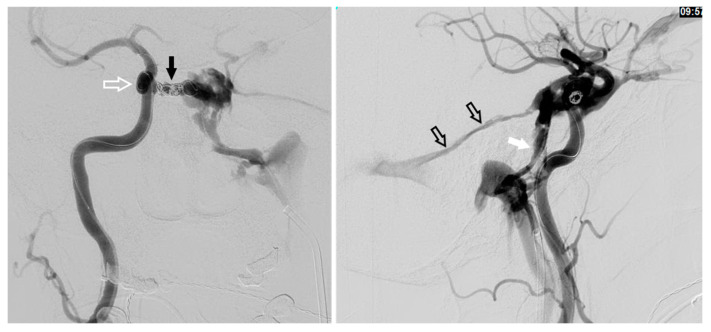
Demonstrates a right sided direct carotid cavernous fistulas (CCF). An occlusion balloon is placed within the right cavernous internal carotid artery (ICA) (white arrow) across the fistula site. Coils are placed within the intercavernous sinus (black solid arrow) through a transvenous inferior petrosal sinus (IPS) approach (white solid arrow). Note the superior petrosal sinus (black arrows), in the lateral projection, with its acute angle off the transverse sinus.

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
