# Peer review of "Endovascular Approaches to the Cavernous Sinus in the Setting of Dural Arteriovenous Fistula"

_brainsci, 2020, doi:10.3390/brainsci10080554_

Round 1

Reviewer 1 Report

In the present study the author review the literature regarding approaches to the CS for endovascular treatments.

Another option if doing a direct superficial stick or cut down and stick of the SOV is to use an angiocath. As the authors note, sometimes the micropuncture wire doesn’t have enough “landing room” before a tight turn. The angiocath shorter than the microdilator. Ultrasound can also be helpful for this.

How did the authors register a spinal needle tip with neuronavigation in the OR? Isite or other programs in the angiography suite also nice for needle trajectory planning for direct puncture SOV/CS.

A few other approaches could be included. Direct Transovale puncture has been reported (Gil et al. JNS).

Endoscopic approach is a straight forward means to access the cavernous sinus – I prefer over craniotomy. This has been published by Tang et al also in JNS.

A few areas need minor editing: “For the IPS approach via the IJV, the authors of the current article prefer to place a 5 Fr 71 sheath in the left common femoral artery (CFA) and a 5 Fr catheter in the common carotid artery 72 (CCA), to be used for diagnostic angiograms and roadmaps.”

Reviewer 2 Report

This is a well written and helpful overview of endovascular approaches to the cavernous sinus. The authors have to be complimented.

As CCF is a rare pathology it might be helpful if the authors could give the readers some insight in how often they have used the various approaches. Do they have a ranking of which approach is their favorite one (-as they mention this is the IPS-)  and do they have a systematic approach in going through the various anatomical options: f.e check IPS first, then SOV etc?

-section 2 IPS: as the authors describe the sheaths and guide catheters they use it would also be helpful to document which microcatheter and microwire they prefer.

-section 4: as this is a trajectory not familiar to most endovascular neurosurgeons it might be helpful to have also a anatomical drawing inserted (like figures 1 and 5). Another option would be to have just one drawing with all the relevant veins and routes visualized.

-figure 9: the image on the right does not add a lot as was am not able to see either Onyx or contrast on that image.

Reviewer 3 Report

The authors presented a concise and comprehensive review of the multiple endovascular transvenous routes to the cavernous sinus (CS) for the treatment of fistula.  The relevant anatomy is reviewed.  More importantly, the manuscript describes in detail the technical approaches used by this highly experienced team to gain access to the CS through the described routes.  The manuscript is well-written and highlights the creativity of endovascular surgeons.  I would certainly refer back to this paper when I encounter difficult cases of CC fistula.  I suspect both the trainees and experienced endovascular surgeons would find this manuscript valuable.   Line 201 – a phrase maybe missing.  I didn’t understand the line “Most of this approach is extracranial and may have a lower risk of subarachnoid hemorrhage, in the event of vessel rupture, than the IPS (‘in-out-in’) route.”   For completeness, direct surgical exposure of cortical vein for endovascular access to the CS has also been described.  The authors may consider including that in the manuscript.
Ghosh R, Al Saiegh F, Mahtabfar A, et al. Burr Hole-Assisted Direct Transsylvian Venous Catheterization for Carotid-Cavernous Fistula Embolization: A Case Report. Oper Neurosurg (Hagerstown). 2020;19(2):E196-E200. doi:10.1093/ons/opz394
